# Correcting for physical distortions in visual stimuli improves reproducibility in zebrafish neuroscience

Timothy W Dunn[1]*, James E Fitzgerald[2]*

[1]Duke Forge, Duke Global Neurosurgery and Neurology, Departments of Statistical Science and Neurosurgery, Duke University, Durham, United States; [2]Janelia Research Campus, Howard Hughes Medical Institute, Ashburn, United States

**Abstract** Optical refraction causes light to bend at interfaces between optical media. This phenomenon can significantly distort visual stimuli presented to aquatic animals in water, yet refraction has often been ignored in the design and interpretation of visual neuroscience experiments. Here we provide a computational tool that transforms between projected and received stimuli in order to detect and control these distortions. The tool considers the most commonly encountered interface geometry, and we show that this and other common configurations produce stereotyped distortions. By correcting these distortions, we reduced discrepancies in the literature concerning stimuli that evoke escape behavior, and we expect this tool will help reconcile other confusing aspects of the literature. This tool also aids experimental design, and we illustrate the dangers that uncorrected stimuli pose to receptive field mapping experiments.

*For correspondence:
timothy.dunn@duke.edu (TWD);
fitzgeraldj@janelia.hhmi.org (JEF)

**Competing interests:** The authors declare that no competing interests exist.

Breakthrough technologies for monitoring and manipulating single-neuron activity provide unprecedented opportunities for whole-brain neuroscience in larval zebrafish (*Ahrens et al., 2012*; *Ahrens et al., 2013*; *Portugues et al., 2014*; *Prevedel et al., 2014*; *Vladimirov et al., 2014*; *Dunn et al., 2016b*; *Naumann et al., 2016*; *Kim et al., 2017*; *Vladimirov et al., 2018*). Understanding the neural mechanisms of visually guided behavior also requires precise stimulus control, but little prior research has accounted for physical distortions that result from refraction and reflection at an air-water interface that usually separates the projected stimulus from the fish (*Sajovic and Levinthal, 1983*; *Stowers et al., 2017*; *Zhang and Arrenberg, 2019*). In a typical zebrafish visual neuroscience experiment, an animal in water gazes at stimuli on a screen separated from the water by a small (~500 μm) region of air (*Figure 1a*, top). When light traveling from the screen reaches the air-water interface, it is refracted according to Snell's law (*Hecht, 2016*; *Figure 1b*, bottom). At flat interfaces, a common configuration used in the literature (*Ahrens et al., 2012*; *Vladimirov et al., 2014*; *Dunn et al., 2016a*), this refraction reduces incident light angles, thereby translating and distorting the images that reach the fish (black vs. brown arrows in *Figure 1a*, bottom). By solving Snell's equations for this arena configuration (Appendix 1), we determined the apparent position of a point on the screen, $\theta$, as a function of its true position, $\theta'$ (*Figure 1b*). Snell's law implies that distant stimuli appear to the fish at the asymptotic value of $\theta(\theta')$ (~48.6°). This implies that the entire horizon is compressed into a 97.2° "Snell window" whose size does not depend on the distances between the fish and the interface ($d_w$) or the screen and the interface ($d_a$), but the distance ratio $d_a/d_w$ determines the abruptness of the $\theta(\theta')$ transformation. We also calculated the total light transmittance according to the Fresnel equations (*Figure 1b*, right). These two effects have a profound impact on visual stimuli (*Figure 1c*). The plastic dish that contains the water has little impact (Appendix 1). Physical distortions thus have the potential to affect fundamental conclusions drawn from studies of visual processing and visuomotor transformations.

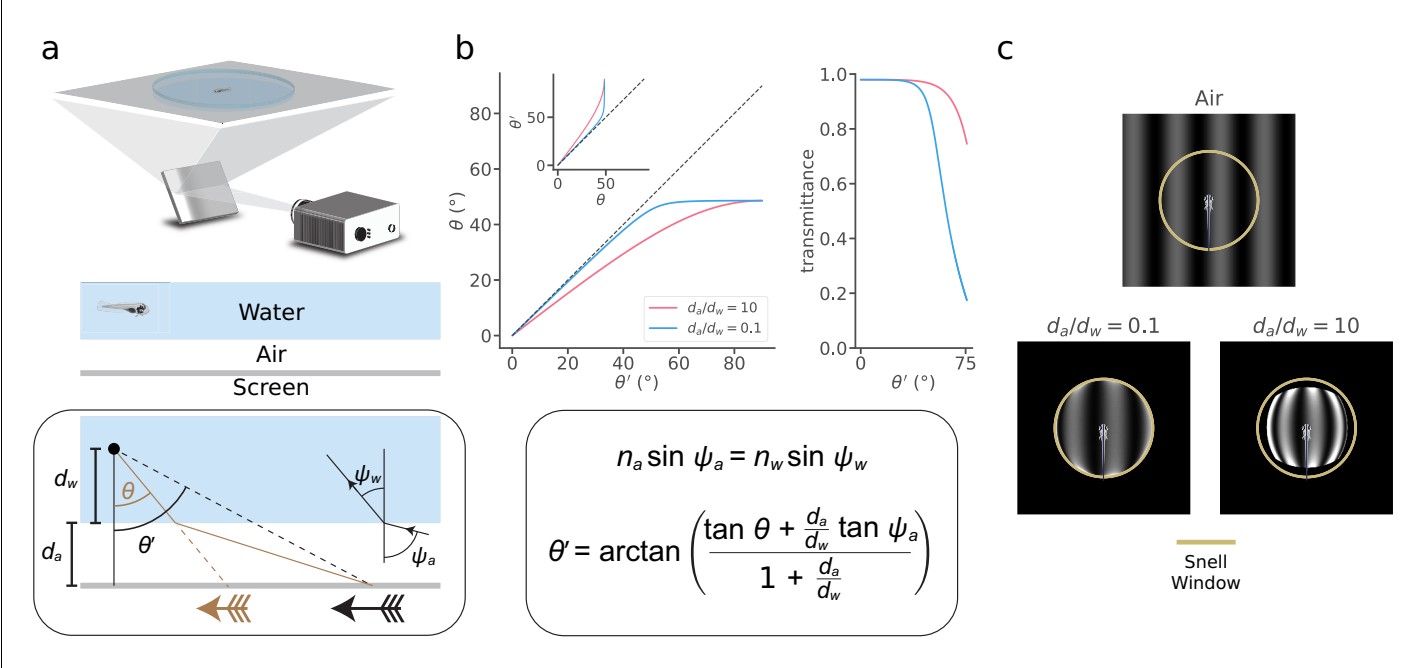

**Figure 1.** Snell's law describes visual stimulus distortions that occur via air-water interfaces encountered in a typical experiment. (**a**) *Top*, In a typical zebrafish neuroscience experiment, an image is presented via projection onto a screen underneath an animal in a water-filled plastic dish. *Middle*, A small layer of air separates the screen from the dish and water. *Bottom box*, This configuration causes the image received at the eye (brown arrow) to be distorted and translated relative to the projected image (black arrow). We can describe this transformation as a relationship between the true position of a projected point ($\theta'$) and its apparent position ($\theta$), depending on the ratio between the distance from the air-water interface to the screen ($d_a$) and the distance from the eye to the air-water interface ($d_w$). To solve the transformation, we use Snell's law (illustrated in inset and panel b), which relates the angle at which a light ray leaves the air-water interface ($\psi_w$) to the angle at which it hits the interface ($\psi_a$), depending on the refractive indices of the media (air, $n_a = 1$; water, $n_w = 1.333$). Note that the effects of the plastic dish are typically minor (Appendix 1). (**b**) *Top left,* the apparent position of a point ($\theta$) as a function of its true position ($\theta'$), and its inverse (*inset*), for $d_a/d_w = 10$ (pink) and $d_a/d_w = 0.1$ (blue). *Top right,* fraction of light transmitted into the water as a function of $\theta'$ for the same two values of $d_a/d_w$. *Bottom box,* Using Snell's law, we derived $\theta'(\theta)$ (*top left inset*), whose inverse we take numerically to arrive at $\theta(\theta')$ (*top left*). (**c**) Simulated distortion of a standard sinusoidal grating. Yellow circle denotes the extent of the Snell window (~97.2° visual angle). The virtual screen is modeled as a 4 × 4 cm square with 250 pixels/cm resolution, and we fixed the total distance between the fish and the virtual screen, $d_a + d_w$, to be 1 cm. Note that only a fraction of the screen is apparent when $d_a/d_w$ is small (*bottom left*), but a distorted view of the full screen appears within the Snell window when $d_a/d_w$ becomes large (*bottom right*). Contrast axes are matched across panels and saturate to de-emphasize the ring of light at the Snell window, whose magnitude would be attenuated by unmodeled optics in the fish eye (Materials and methods).

The quantitative merits of correcting for refraction are apparent when comparing two recent studies of visually evoked escape behavior in larval zebrafish. Although *Temizer et al. (2015)* and *Dunn et al., 2016a* both found that a critical size of looming stimuli triggered escape behavior, they reported surprisingly different values for the critical angular size (21.7°±4.9° and 72.0°±2.5°, respectively, mean ±95% CI). This naively implies that the critical stimulus of Dunn et al. occupied 9 times the solid angle of Temizer et al. (1.02 [+0.14,–0.11] steradians and 0.11 [+0.06,–0.04] steradians, respectively, mean [95% CI]) (Materials and methods). This large size discrepancy initially raises doubt to the notion that a stimulus size threshold triggers the escape (*Hatsopoulos et al., 1995*; *Gabbiani et al., 1999*; *Fotowat and Gabbiani, 2011*). However, a major difference in experimental design is that Temizer et al. showed stimuli from the front through a curved air-water interface, and Dunn et al. showed stimuli from below through a flat air-water interface (*Figure 2a*). Correcting the Dunn et al. stimuli with Snell's law, and again quantifying the size of irregularly shaped stimuli with their solid angle, we found that the fish exhibited escape responses when the stimulus spanned just 0.24 steradians (*Figure 2b*, Materials and methods, Appendix 1, *Figure 2—video 1*). The same correction applied to Temizer et al. sets the critical size at 0.08 steradians (*Figure 2b*, Materials and methods, Appendix 2). This leaves a discrepancy of 0.16 steradians, which is much smaller than the original solid angle discrepancy of 0.91 steradians (*Figure 2c*, *black*). Correcting with Snell's law thus

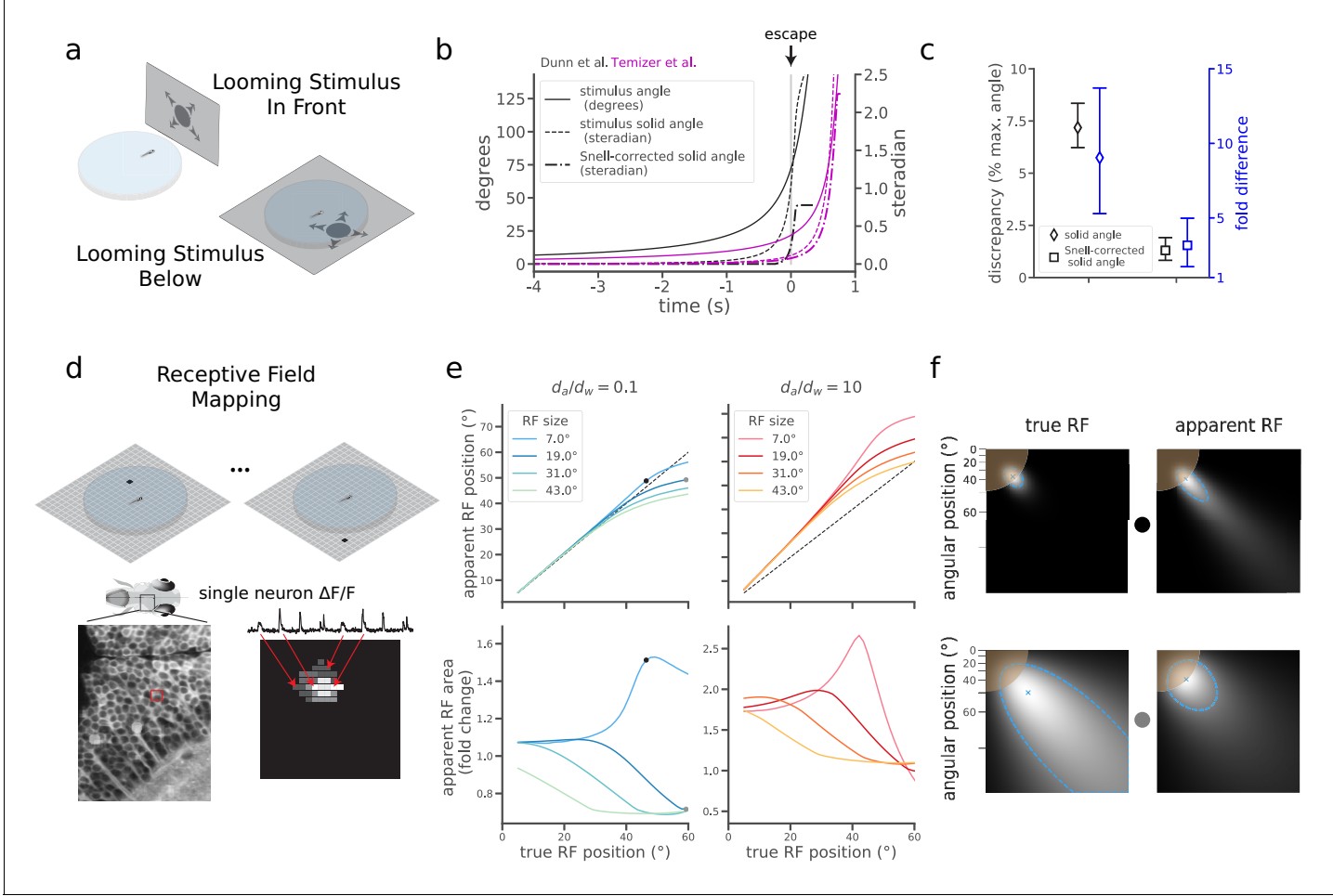

**Figure 2.** Snell's law corrections reduce discrepancies in the literature and predict effects on receptive field mapping. (a) In the zebrafish literature, two configurations were used to probe the neural circuitry processing looming stimuli that expand over time. In one, fish were embedded off-center in a curved plastic dish and a screen presented stimuli in front of the animal through the curved interface of the dish (*Temizer et al., 2015*). In the other, fish were embedded (or swam freely) in a similar dish, but stimuli were presented on a screen below the dish (as in *Figure 1a*; *Dunn et al., 2016a*). (b) Plot detailing the changes to the looming expansion time courses after correcting for Snell's law and converting to solid angle, which more accurately describes the irregular stimulus shapes produced by the optical distortion (Materials and methods). Curves corresponding to Dunn et al. and Temizer et al. are plotted in black and magenta, respectively. (c) Snell's law corrections reduced the discrepancy between Dunn et al. and Temizer et al. *Black*: Snell's law corrections decreased the absolute magnitude of the discrepancy (Dunn et al. critical solid angle minus Temizer et al. critical solid angle). We report discrepancies as fractions of the maximal solid angle (4π steradians) to aid intuition for stimulus sizes. *Blue*: Snell's law corrections also decreased the relative magnitude of the discrepancy (Dunn et al. size divided by Temizer et al. size). (d) In a simple receptive field (RF) mapping experiment, dots appear at different positions on a screen (*Top*), and behavioral or neural responses (*Bottom*) are measured. In the latter case, a map of a single neuron's RF is constructed by assigning the measured $\Delta F/F$ signal to the point on the screen that evoked the $\Delta F/F$ response. (e) Snell's law predicts changes in RF peak positions (*Top*) and RF sizes (*Bottom*). The magnitude of these changes depends on the true RF position (x-axis), true RF size (line color), and $d_a/d_w$ (warm versus cool colors). True RF positions and sizes correspond to the means and standard deviations of Gaussian receptive fields. The black dots indicate the RFs in *panel f, top*, and the gray dots show the RFs in *panel f, bottom*. (f) Illustrations of two simulated "true" RFs and their corresponding measurement distortions predicted using Snell's law. For simplicity, we show only one quadrant of the screen space, with the fish at the top left corner. The brown circle denotes the extent of the Snell window. As RFs are mapped directly to screen pixels, the axes are nonlinear in terms of angle relative to the fish (top left corner). Each blue "x" denotes the peak position of the RF displayed in each plot. The dashed blue border denotes the half-maximum value of each RF, and the size of the RF is the solid angle within one of these borders.

The online version of this article includes the following video for figure 2:

**Figure 2—video 1.** Optical refraction distorts the appearance of looming visual stimuli.

https://elifesciences.org/articles/53684#fig2video1

markedly reduced this discrepancy in the literature, shrinking a 9-fold size difference down to 3-fold (*Figure 2c*, *blue*). The small remaining difference could indicate an ethologically interesting dependence of behavior on the spatial location of the looming stimulus (*Dunn et al., 2016a*; *Temizer et al., 2015*).

Accounting for optical distortions will be critical for understanding other fundamental properties of the zebrafish visual system. For example, a basic property of many visual neurons is that they respond strongest to stimuli presented in one specific region of the visual field, termed their receptive field (RF) (*Hartline, 1938*; *Ringach, 2004*; *Zhang and Arrenberg, 2019*). When we simulated the effect of Snell's law on RF mapping under typical experimental conditions (*Figure 2d*), we predicted substantial errors in both the position and size of naively measured receptive fields (*Figure 2e*, Materials and methods). Depending on the properties of the true RF, its position and size could be either over- or under-estimated (*Figure 2e–f*), with the most drastic errors occurring for small RFs appearing near the edge of the Snell window.

Future experiments could avoid distortions altogether by adjusting experimental hardware. For instance, fish could be immobilized in the center of water-filled spheres (*Zhang and Arrenberg, 2019*; *Dehmelt et al., 2019*), or air interfaces could be removed altogether, such as by placing a projection screen inside the water-filled arena. But in practice the former would restrict naturalistic behavior, and the latter would reduce light diffusion by shrinking the refractive index mismatch between the diffuser and transparent medium (water vs. air) that typical light diffusers use to transmit stimuli over a large range of angles. An engineering solution might build diffusive elements into the body of the fish tank (*Stowers et al., 2017*; *Franke et al., 2019*). Alternatively, we propose a simple computational solution to account for expected distortions when designing stimuli or analyzing data. Our tool (https://www.github.com/spoonsso/snell_tool/) converts between normal and distorted image representations for the most common zebrafish experiment configuration (*Figure 1a*), and other geometries could be analyzed similarly. This tool will therefore improve the interpretability and reproducibility of innovative experiments that capitalize on the unique experimental capabilities available in zebrafish neuroscience.

## Materials and methods

See Appendix 1 and Appendix 2 for the geometric consequences of Snell's law at flat and curved interfaces, respectively.

### Implications of the Fresnel equations

Only a portion of the incident light is transmitted into the water to reach the eye. We calculated the fraction of transmitted light according to the Fresnel equations. Assuming the light is unpolarized,

$$T = 1 - \frac{R_s + R_p}{2},$$

where $T$ is the fraction of light transmitted across an air-water interface at incident angle $\psi_a = \psi_a(\theta)$ (See Appendices 1, 2), $\psi_w = \theta$ is the angle of the refracted light ray in water, and

$$R_s = \left( \frac{n_a \cos \psi_a(\theta) - n_w \cos \theta}{n_a \cos \psi_a(\theta) + n_w \cos \theta} \right)^2$$
$$R_p = \left( \frac{n_a \cos \theta - n_w \cos \psi_a(\theta)}{n_a \cos \theta + n_w \cos \psi_a(\theta)} \right)^2$$

are the reflectances for s-polarized (i.e. perpendicular) and p-polarized (i.e. parallel) light, respectively. When including the plastic dish in our simulations, we modified these equations to separately calculate the transmission fractions across the air-plastic and the plastic-water interfaces. We assumed that the full transmission fraction is the product of these two factors, thereby ignoring the possibility of multiple reflections within the plastic.

### Illustrating distorted sinusoidal gratings

For all image simulations in *Figure 1c*, we neglected the plastic and fixed the total distance between the fish and the virtual screen, $d_a + d_w$, to be 1 cm, a typical distance in real-world experiments. The virtual screen was considered to be a 4 × 4 cm square with 250 pixels/cm resolution. Here we

assumed that the virtual screen emits light uniformly at all angles, but this assumption is violated by certain displays, and our computational tool allows the user to specify alternate angular emission profiles. To transform images on the virtual screen, we shifted each light ray (i.e. image pixel) according to Snell's law, scaled its intensity according to the Fresnel equations, and added the intensity value to a bin at the resulting apparent position. This simple model treats the fish eye as a pinhole detector, whereas real photoreceptors blur visual signals on a spatial scale determined by their receptive field. Consequently, our simulation compresses a large amount of light onto the overly thin border of the Snell window, and we saturated the grayscale color axes in *Figure 1c* to avoid this visually distracting artifact.

To make the image as realistic as possible, we mimicked real projector conditions using gamma-encoded gratings with spatial frequency 1 cycle / cm, such that

$$[x(t)]^{1/2.2} = \sin t$$

with $x(t)$ ranging from 1.0 to 500.0 lux, a standard range of physical illuminance for a lab projector. The exponent on the left represents a typical display gamma encoding with gamma = 2.2. To reduce moiré artifacts arising from ray tracing, we used a combination of ray supersampling (averaging the rays emanating from 16 sub-pixels for each virtual screen pixel) and stochastic sampling (the position of each ray was randomly jittered between -1 and 1 sub-pixels from its native position) (*Dippé and Wold, 1985*). In *Figure 1c*, we display the result of these operations followed by a gamma compression to mimic the perceptual encoding of the presented stimulus.

## Corrections to looming visual stimuli

We approximated the geometric parameters from *Dunn et al. (2016a)* (flat air-water interface, $d_a$ = 0.5 mm, $d_w$ = 3 mm, $d_p$ = 1 mm, stimulus offset from the fish by 10 mm along the screen) and *Temizer et al., 2015* (curved air-water interface, $d_a$ = 8 mm, $d_w$ = 2 mm, $d_p$ = 1 mm, r = 17.5 mm, stimulus centered) to create Snell-transformed images of circular stimuli with sizes growing over time (*Figure 2a–c*). We used a refractive index of $n_p$ = 1.55 for the polystyrene plastic. While Dunn et al. collected data from freely swimming fish, the height of the water was kept at approximately 5 mm, and 3 mm reflects a typical swim depth. Since freely swimming zebrafish can adjust their depth in water, it's an approximation to treat $d_w$ as constant.

We quantified the size of each transformed stimulus with its solid angle, the surface area of the stimulus shape projected onto the unit sphere. To calculate the solid angle for Temizer et al., we used the formula for a spherical cap, $A = 2\pi(1 - \cos\theta)$, where $A$ is the solid angle and $2\theta$ is the apex angle. To calculate the solid angle for Dunn et al., in which stimuli were not spherical caps, we first represented stimulus border pixels in a spherical coordinate system locating the fish at the origin. The radial coordinate does not affect the solid angle, so we described each border pixel by two angles: the latitude, $\alpha$, and longitude, $\beta$. To calculate the area, we used an equal-area sinusoidal (Mercator) projection given by

$$(x, y) = (\beta\cos\alpha, \alpha),$$

which projects an arbitrary shape on the surface of a sphere onto the Cartesian plane. While distances and shapes are not preserved in this projection, area as a fraction of the sphere's surface area is maintained. Thus, we could calculate the solid area of the stimulus in this projection by finding the area of the projected 2D polygon. To calculate the absolute and relative discrepancy 95% confidence intervals in *Figure 2c*, we used error propagation formulae for the difference and division of two distributions, respectively.

## Receptive field mapping

We simulated receptive field (RF) mapping experiments by tracing light paths from single pixels on a virtual screen to the fish (*Figure 2d-f*). We modeled a neuron's RF as a Gaussian function on the sphere, defined the "true RF" to be the pixel-wise response pattern that would occur in the absence of the air-water interface, and defined the "apparent RF" as the pixel-wise response pattern that would be induced with light that bends according to Snell's law at an air-water interface. More precisely, we modeled the neural response to pixel activation at position $x$ as

$$F(x) = T(\psi_a(x)) P(\rho(x, \mu_{RF}), \sigma_{RF}^2),$$

where $T(\psi_a(x))$ is the fraction of light transmitted (Fresnel equations), $\mu_{RF}$ and $\sigma_{RF}$ are the mean and standard deviation of the Gaussian RF, $\rho(x, \mu_{RF})$ is the distance along a great circle from the center of the RF to the pixel's projected retinal location, and $P(\rho, \sigma_{RF}^2) = e^{-\rho^2/(2\sigma_{RF}^2)}$ is the Gaussian RF shape. We calculated the great circle distance between points on the sphere as

$$\cos \rho(x, \mu_{RF}) = \sin \alpha_{RF} \sin \alpha_x + \cos \alpha_{RF} \cos \alpha_x \cos(\beta_x - \beta_{RF}),$$

where $(\alpha_{RF}, \beta_{RF})$ are the latitude and longitude coordinate of the RF center, and $(\alpha_x, \beta_x)$ are the latitude and longitude coordinates of the projected pixel location. We quantified the position of the RF as the maximum of $F(x)$, converted to an angular coordinate along the screen. We quantified RF area as the solid angle of the shape formed by thresholding $F(x)$ at half its maximal value.

## Computational tool for simulating and correcting optical distortions

With this paper, we provide a computation tool for visualizing and correcting distortions (https://github.com/spoonsso/snell_tool/). The tool is written in Python and uses standard image processing libraries. The tool can be launched virtually over the web, without any need to install new software, using the MyBinder link in the README file hosted on the github repository. The source code can also be downloaded and run on the user's local machine.

The uses and parameters of the tool are described in detail in an example notebook in the repository (*snell_example.ipynb*). In brief, the tool is implemented only for flat interfaces with the assumptions described in Appendix 1, and it can model distortions through three media (i.e. with a plastic interface between air and water). It can also model displays that emit light with non-uniform angular profiles. Key customizable parameters include the screen size, screen resolution, screen distance, media thicknesses, media refractive indices, and gamma encoding. As described in Illustrating distorted sinusoidal gratings, the tool uses a combination of ray super-sampling and stochastic sampling to reduce moiré artifacts arising from ray tracing.

The Python notebook illustrates two primary use cases of the tool, though the tool's library is flexible enough to be adopted for other tasks. First, it allows the user to input an image to see its distorted form under the assumptions of the model. Thus, it recreates *Figure 1c*, but for any arbitrary grayscale stimulus, and for a range of user-specified experimental configurations. Second, it allows the user to input an undistorted target image, and the tool inverts the distortion process to suggest an image that could be displayed during an experiment to approximately produce the target from the point of view of the fish. In the tool's example notebook, we demonstrate this inversion process using a checkered ball stimulus. Importantly, note that some stimuli will be physically impossible to correct (e.g. undistorted image content cannot be delivered outside the Snell window).

## Acknowledgements

We thank Damon Clark, Ruben Portugues, and Kristen Severi for helpful comments on the manuscript. We thank Eva Naumann for discussions regarding light diffusion in the laboratory and for sharing fish icons for the figures. We also thank Florian Engert and Haim Sompolinsky for early support and partial funding of the project (NIH grant U01 NS090449). TWD was supported by Duke Forge and Duke AI Health. JEF was supported by the Howard Hughes Medical Institute.

## Additional information

### Funding

| Funder | Grant reference number | Author |
| --- | --- | --- |
| Duke Forge | | Timothy W Dunn |
| Duke AI Health | | Timothy W Dunn |
| Howard Hughes Medical Institute | | James E Fitzgerald |

| National Institutes of Health | U01 NS090449 | Timothy W Dunn James E Fitzgerald |
| --- | --- | --- |

The funders had no role in study design, data collection and interpretation, or the decision to submit the work for publication.

## Author contributions

Timothy W Dunn, Conceptualization, Software, Formal analysis, Funding acquisition, Validation, Investigation, Visualization, Methodology, Project administration, Writing – original draft, Writing – review and editing; James E Fitzgerald, Conceptualization, Formal analysis, Funding acquisition, Validation, Investigation, Methodology, Project administration, Writing – original draft, Writing – review and editing

## Author ORCIDs

Timothy W Dunn (iD) https://orcid.org/0000-0002-9381-4630
James E Fitzgerald (iD) https://orcid.org/0000-0002-0949-4188

## Decision letter and Author response

Decision letter https://doi.org/10.7554/eLife.53684.sa1
Author response https://doi.org/10.7554/eLife.53684.sa2

# Additional files

## Supplementary files

• Transparent reporting form

## Data availability

No data were collected for this theoretical manuscript.

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

## Appendix 1

### Implications of Snell's law at a flat interface

For this and all subsequent analyses, we treat the fish as a pinhole detector. Here we derive $\theta'(\theta)$ with the aid of **Appendix 1—figure 1**. Note that this derivation includes optical effects from the plastic dish, but these effects will be relatively minor. To begin, we summarize the basic trigonometry of the problem. The true angular position of the stimulus is given by

$$\tan\theta' = \frac{d'_w + d'_p + d'_a}{d_w + d_p + d_a},$$

where $d_w$ is the normal distance between the fish and the water-plastic interface, $d_p$ is the normal distance between the water-plastic and plastic-air interfaces, $d_a$ is the normal distance between the air interface and the screen (interface and screen assumed to be parallel), $d'_w$ is the parallel distance traveled by the light ray in the water, $d'_p$ is the parallel distance traveled by the light ray in the plastic, and $d'_a$ is the parallel distance traveled by the light ray in air. Each parallel distance is related to the corresponding normal distance by simple trigonometry. The apparent angular location of the stimulus satisfies

$$d'_w = d_w \tan\theta,$$

and the incident light angle satisfies

$$\begin{aligned} d'_a &= d_a \tan\psi_a, \\ d'_p &= d_p \tan\psi_p, \end{aligned}$$

thereby leading to

$$\theta' = \tan^{-1}\left(\frac{d_w\tan\theta + d_p\tan\psi_p + d_a\tan\psi_a}{d_w + d_p + d_a}\right).$$

We can next use Snell's law to reduce the number of angular variables. In particular,

$$\psi_p = \sin^{-1}\left(\frac{n_w\sin\psi_w}{n_p}\right) = \sin^{-1}\left(\frac{n_w\sin\theta}{n_p}\right)$$

and

$$\psi_a = \sin^{-1}\left(\frac{n_p\sin\psi_p}{n_a}\right) = \sin^{-1}\left(\frac{n_p\sin\left(\sin^{-1}\left(\frac{n_w\sin\theta}{n_p}\right)\right)}{n_a}\right) = \sin^{-1}\left(\frac{n_w\sin\theta}{n_a}\right)$$

together imply that

$$\theta'(\theta) = \tan^{-1}\left(\frac{d_w\tan\theta + d_p\tan\left(\sin^{-1}\left(\frac{n_w\sin\theta}{n_p}\right)\right) + d_a\tan\left(\sin^{-1}\left(\frac{n_w\sin\theta}{n_a}\right)\right)}{d_w + d_p + d_a}\right).$$

The role of plastic in this equation is typically minimal. To see this, first note that $n_w \approx 1.333 < n_p \approx 1.55$, which implies that $\frac{n_w\sin\theta}{n_p} < 1$. This implies that the Snell window is determined by $1 = \frac{n_w\sin\theta}{n_a}$, and the properties of the high-index plastic dishes have no effect on the size of the Snell window. The plastic can cause distortions within the Snell window, but these effects were small for all experimental arenas analyzed in this paper, as we empirically found that none of our results qualitatively depended upon the plastic. We therefore chose to highlight the critical impact of the air-water interface by assuming that $d_p = 0$ in the main text's conceptual discussion. We nevertheless included nonzero values of $d_p$ in our computational tool so that users can account for the quantitative effects of the plastic dish. We also included the effects of plastic when quantitatively correcting previously published results. Because

analytically inverting $\theta'(\theta)$ is non-trivial, we noted from the graph of $\theta'(\theta)$ that the inverse function exists and calculated $\theta(\theta')$ with a numerical look-up table (*e.g.* **Figure 1b**).

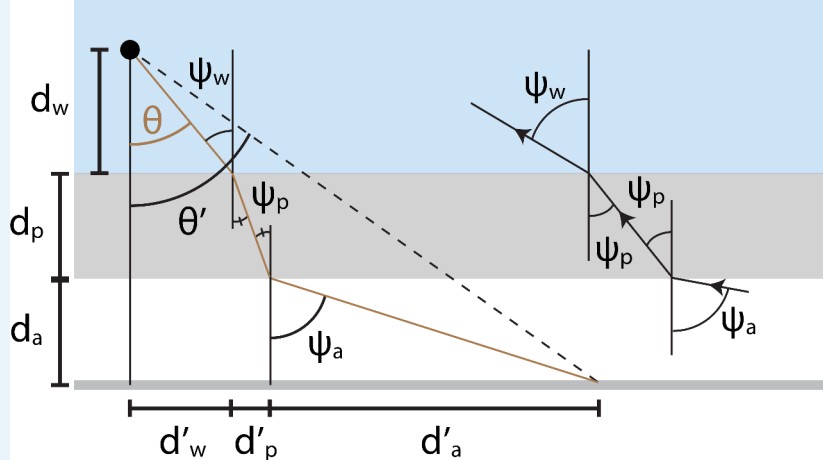

**Appendix 1—figure 1.** Illustration of mathematical variables used to analyze optical distortions in arena geometries where flat air-plastic and plastic-water interfaces separate the fish from the projection screen. The brown line denotes the trajectory of a light ray traveling from the screen to the fish. We quantify the image transformation by relating the true angular position of each projected point ($\theta'$) to its apparent position ($\theta$). The derivation involves several distances (e.g. $d_w$) that summarize the ray's trajectory through air (white region), plastic (gray region), and water (blue region). Refraction angles ($\psi_a, \psi_p, \psi_w$) describe the bending of light at each interface.

## Appendix 2

### Implications of Snell's law at a curved interface

When the fish is mounted off-center (*Appendix 2—figure 1a*) in a circular dish (brown dot), rays pass through a curved interface and are refracted at tangent lines (brown line). We begin by using Snell's law and basic trigonometry to relate each refraction angle to $\theta$. Let $d_a$ denote the distance in air between the edge of the plastic dish and the screen, $d_p$ denote the thickness of the plastic dish, $d_w$ denote the distance in water between the fish and the edge of the tank nearest the screen, and $r$ denote the radius of the dish (excluding the plastic). We assume that $d_w \leq r$ and the screen is perpendicular to the line between the fish and the center of the dish. Cases where the fish is behind the dish's center or the screen is angled can be analyzed similarly. Starting at the fish and moving outwards, we first apply the law of sines to the gray triangle to find

$$\frac{\sin(\psi_w)}{r - d_w} = \frac{\sin(\pi - \theta)}{r} \Longrightarrow \sin(\psi_w(\theta)) = \frac{r - d_w}{r}\sin(\theta),$$

where we've used the identity $\sin(\pi - x) = \sin(x)$. It will be useful for later to note that this triangle also implies that $\gamma = \pi - (\psi_w + \pi - \theta) = \theta - \psi_w$. Snell's law at the plastic-water interface implies,

$$n_p \sin(\psi_p) = n_w \sin(\psi_w) \Longrightarrow \sin(\psi_p(\theta)) = \frac{n_w}{n_p}\sin(\psi_w) = \frac{n_w}{n_p}\frac{r - d_w}{r}\sin(\theta).$$

We next relate the two plastic refraction angles to each other by applying the law of sines to the orange triangle and find

$$\frac{\sin(\psi_p')}{r} = \frac{\sin(\pi - \psi_p)}{r + d_p} \Longrightarrow \sin(\psi_p'(\theta)) = \frac{r}{r + d_p}\sin(\psi_p) = \frac{n_w}{n_p}\frac{r - d_w}{r + d_p}\sin(\theta).$$

Finally, we determine the dependence of $\psi_a$ on $\theta$ from Snell's law applied to the air-plastic interface,

$$n_a \sin(\psi_a) = n_p \sin(\psi_p') \Longrightarrow \sin(\psi_a(\theta)) = \frac{n_p}{n_a}\sin(\psi_p') = \frac{n_w}{n_a}\frac{r - d_w}{r + d_p}\sin(\theta).$$

With these formulae in hand, we now proceed to the main goal of deriving an expression for $\theta'(\theta)$. Since we've already extracted everything from Snell's law, all that remains is basic trigonometry, which we illustrate in *Appendix 2—figure 1b*. First note that applying the definition of the tangent function to the blue triangle implies that

$$\theta'(\theta) = \tan^{-1}\left(\frac{s(\theta) + s'(\theta)}{d_a + d_p + d_w}\right).$$

It thus suffices to determine expressions for $s(\theta)$ and $s'(\theta)$. Consider first $s'(\theta)$. The large red triangle implies

$$s'(\theta) = (r + d_p + d_a)\tan(\alpha(\theta)).$$

Rewriting $\alpha$ in terms of the other two angles in the $\alpha\beta\psi_p'$ triangle gives $\alpha = \pi - \beta - \psi_p'$. Rewriting $\beta$ in terms of the other two angles in the $\beta\gamma\psi_p$ triangle gives $\beta = \pi - (\gamma + \psi_p) = \pi - \theta + \psi_w - \psi_p$. Putting these pieces together, we thus find

$$\alpha(\theta) = \theta - \psi_p'(\theta) + \psi_p(\theta) - \psi_w(\theta).$$

Next consider $s(\theta)$. Applying the law of sines to the green triangle, we find

$$\frac{\sin(\psi_a)}{s} = \frac{\sin(\omega)}{a} \Rightarrow s(\theta) = \frac{a(\theta)\sin(\psi_a(\theta))}{\sin(\omega(\theta))}.$$

Rewriting $\omega$ in terms of the other two angles in the green triangle, $\omega = \pi - (\psi_a + \pi - \varphi) = \varphi - \psi_a$, and rewriting $\varphi$ in terms of the other angles in the red triangle, $\varphi = \pi - \left(\alpha + \frac{\pi}{2}\right) = \frac{\pi}{2} - \alpha$, we find

$$\omega(\theta) = \frac{\pi}{2} - \theta - \psi_a(\theta) + \psi'_p(\theta) - \psi_p(\theta) + \psi_w(\theta).$$

Finally, we find the dependence of $a$ on $\theta$ from the red triangle using the definition of the cosine function

$$\cos(\alpha) = \frac{r + d_p + d_a}{r + d_p + a} \Longrightarrow a(\theta) = \frac{r + d_p + d_a}{\cos(\alpha(\theta))} - r - d_p.$$

Since we've written $\alpha$, $a$, $\omega$, and the refraction angles as functions of $\theta$, we've fully specified $s(\theta)$, $s'(\theta)$, and thus $\theta'(\theta)$. As with the flat interface, we calculated $\theta(\theta')$ using a numerical look-up table.

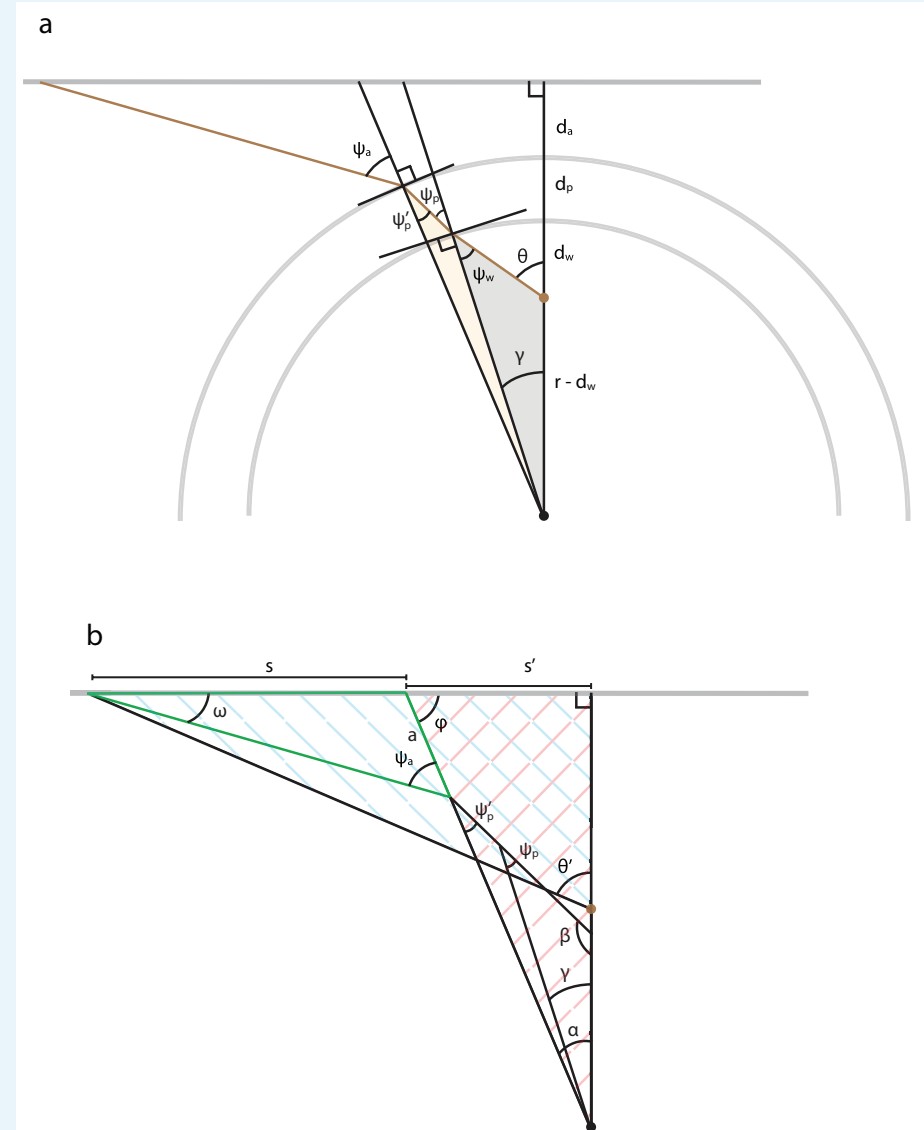

**Appendix 2—figure 1.** Illustration of mathematical variables used to analyze optical refraction at curved interfaces. (**a**) We assume that the interfaces are circular, that the fish is mounted off-center (brown dot), and that the screen and fish are at the same elevation. We neglect distortions that could result from the flat vertical interface running parallel to the longitudinal axis of the cylindrical dish. We denote the radius of the arena's water-filled compartment as $r$. The derivation additionally involves several distances that summarize the placement of the fish in the dish ($d_w$), the thickness of the plastic ($d_p$), and the distance separating the dish from the screen ($d_a$). Refraction angles ($\psi_a, \psi'_p, \psi_p, \psi_w$) of the light ray (brown line) are relative to each interface's normal vector and describe the bending of light. Each shaded region highlights a triangle whose trigonometry is helpful for relating the refraction angles to the apparent angular position of a light source ($\theta$). (**b**) Illustration of mathematical variables used to trigonometrically relate the true angular position of each projected point ($\theta'$) to its apparent position ($\theta$), assuming the same arena geometry as panel $a$. The derivation utilizes most triangles shown, several of which are cross-hatched or outlined to direct the reader's eye.

