## [Decision Letter]

Thank you for submitting your article "Correcting for physical distortions in visual stimuli improves reproducibility in zebrafish neuroscience." for consideration by *eLife*. Your article has been reviewed by three peer reviewers, and the evaluation has been overseen by a Reviewing Editor and Didier Stainier as the Senior Editor. The following individual involved in review of your submission has agreed to reveal their identity: Andre Maia Chagas (Reviewer #3).

The reviewers have discussed the reviews with one another and the Reviewing Editor has drafted this decision to help you prepare a revised submission.

Summary:

The three reviewers agree that the study is of general interest and will help future studies on visual integration in zebrafish properly design their setups. Precisions and corrections are listed below. Please address all points raised and improve clarity of figures, check legends of supplementary videos.

Reviewer #1:

In this manuscript, the authors describe how to correct for distortions of the light path in vision experiments in aquatic animals. They first discuss and illustrate relevant physical laws and then showcase how distortions associated with suboptimal stimulus configurations could have affected two recent studies on looming stimuli in zebrafish. The impact of their manuscript can be subdivided into the following four points.

1) They raise awareness about stimulus stretching, compression and attenuation occurring in aquatic stimulus setups, which have oftentimes been ignored in zebrafish vision research. This will certainly be of high interest for the growing community of scientists working with aquatic animals, especially zebrafish.

2) The authors show that a critical parameter (d_a_/d_w_) determines how the physical laws (Fresnel equations, Snell's law) will affect the stimulus appearance in one of the frequently used recording configurations: the apparent position of a stimulus is shown to depend on the ratio of the water column length and the free-air distance between display and water container (d_a_/d_w_). While the physical laws are no news, the provided equation can be useful for scientists working with such a recording configuration.

3) They calculate, how stimuli have likely been distorted in two recent publications. This is a very helpful type of analysis that can potentially correct/explain observed differences in studies. The authors state that in the original publications the critical stimulus sizes were reported as 21.7° and 72.0° (difference: factor 3.3), and after their correction, the stimulus sizes were 0.6% and 1.9% (of 4π steradians, difference: factor 3.2). If I understand the authors correctly, the authors take the small *absolute* difference of 1.9% and 0.6% ( = 1.3%) as evidence for a better match after optical correction, because the absolute difference (14%) in the original studies before correction (72° – 21.7°=50.3° ~ = 14% of 360 degrees) had been much bigger. However, I don't agree that this is the best way to compare the results. The *relative* difference is about a factor of 3 in both cases, so not much different after correction. That being said, it is of course nonetheless very useful to perform these distortion calculations as they can be used to report the stimulus characteristics from the animal's perspective, which is what matters.

4) A programming tool is provided to help scientists calculate the distortions in their setups. It would be helpful if the functionality (inputs, outputs, available configurations) were described in more detail in the manuscript. The tool can be used for the experimental setup in Figure 1A, but for other configurations, users likely need to adapt it or use other software for ray tracing.

In summary, I think that the topic of the manuscript will be of interest to the community of aquatic vision researchers. The presented results and discussion present a rather incremental scientific advance, but are timely for zebrafish vision researchers.

Reviewer #2:

In this study, Dunn and Fitzgerald evaluate the effects of refractive index mismatches and consequent visual distortions upon zebrafish visual neuroscience experiments. They show that refraction at water-air interfaces causes distortions and translations to visual stimuli that, if ignored, will lead to erroneous conclusions about receptive field properties of visual neurons and visuomotor relationships. They show that accounting for Snell's law explains a large part of the apparent discrepancy between the critical angular sizes reported in two studies of looming-evoked escape behavior. The authors also provide a software tool to simulate the effects of refraction on visual stimuli.

Overall, this paper and the associated software will be of considerable value to the growing community of researchers investigating the larval zebrafish visual system. By highlighting an issue that has hitherto largely been ignored and providing means to improve experimental design and interpretation, the paper should enhance the quality and reproducibility of experiments in this important neuroscience model.

I have two main comments:

1) The authors do not appear to have considered the angular intensity profiles of the screens used to present visual cues. However, this will influence the most eccentric location at which an image feature (pixel) is visible to the fish as well as the observed intensity distribution. Moreover, the types of screens used in different studies vary. For example, Dunn et al. used a diffusive screen (for back projection) which will have some scattering characteristic – how was this modeled? Temizer et al. used an OLED screen where individual pixels have a (likely more limited) angular emission profile. This will determine the maximal theta' from which a ray can be emitted that will reach the fish, causing clipping at eccentric locations an affecting the size and amplitude profile of receptive fields. Can the authors incorporate the angular intensity profile of the presentation screen into their modeling (tool)?

2) It was not clear to me why the maximum image size is assumed to be 360° or 4π steradians. In the Dunn and Temizer papers, I assume angular size is computed according to a simple 2D model where one eye views a directly approaching orthogonal object.

In this case, the largest angular size subtended at the eye is 180° (at impact). However, the largest image size cast on the retina of a real eye will be limited by the visual field of that eye (and thus occur prior to impact). Similarly, for the analysis of solid angle, the authors compute percentages assuming a maximal size of 4π, which is clearly not biologically plausible. I would find it more useful if image sizes were given as a proportion of the (monocular) visual field. What is this estimated to be?

Reviewer #3:

I would like to start by thanking the authors for putting this paper together, as it is well written and definitely an asset for experimenters working with aquatic species, or in general for any other researchers that have visual stimuli that need to go through different media before reaching the animal/subject. I'm surprised that this issue hasn't been raised before, but glad to see that this is done here. I also enjoy the fact that they made a repository with more details on the tool they are proposing as a solution to the issue reported here, also that you can reproduce the paper's figures using this repository.

Here are some comments I hope will make the paper even more enjoyable:

1) Would it be possible for the authors to setup their repository to work with "My Binder" (https://mybinder.org/), so that readers who are not using/familiar with python could still use their notebook?

2) In the second paragraph of the main text the authors state that in traditional experimental setups for zebrafish, due to Snell's law, stimuli that are distant appear to the fish at the asymptotic value of ~48.6°, and that this would lead to a "Snell window" of 97.2°. I think this leads to some interesting questions:

2.1) In this setup configuration, the Snell window is covering what area of the bottom of the Petri dish? It would be nice to see visually represented what is the projection size of the window as size of the dish, since:

2.1.1) If the window is only covering part of the dish, the animals are freely moving, and the correction algorithm is used, this would mean that as the animals move, they could go past the edge of the window and be in a region where the stimulus is not present?

2.1.2) It would be nice to know how big the window projection is in relation to the animal's visual space, and how the distance between the petri dish and the projection window change this

3) The authors also mention that having water instead of air in the space between the petri-dish and the projection could be used as an alternative solution to the problem (main text, last paragraph), with the shortcoming of having a reduced transmission of stimuli in large angles. I wonder what these large angles would be? Would this be something easy to calculate? I wonder about this, since Franke et al. (https://elifesciences.org/articles/48779), show a "fish cinema" system where a lot of the fish's field of view is covered (while projecting directly at a screen which was the water container wall – so no air or water interface before the projected image)

4) One thing that I didn't see mentioned in the paper is chromatic aberration. Given the fact that stimuli are travelling through 3 different media at least, one could expect that different wavelengths will be projected slightly misaligned when compared to one another, especially considering the wide chromatic range present in Zebrafish vision? I wonder if it would be easy to incorporate these calculations in the tools they are describing, and most importantly, how could this affect the calculation of receptive fields, considering the misaligned patches of different wavelengths could directly influence some of these receptive fields?

I'm happy to clarify any points that might be unclear, or provide further arguments for the above mentioned points.

[Editors' note: further revisions were suggested prior to acceptance, as described below.]

Thank you for resubmitting your work entitled "Correcting for physical distortions in visual stimuli improves reproducibility in zebrafish neuroscience." for further consideration by *eLife*. Your revised article has been evaluated by Didier Stainier as the Senior Editor, and a Reviewing Editor.

The manuscript has been improved but there are some remaining issues that need to be addressed before acceptance, as outlined below:

The reviewers have addressed most of the comments from the three reviewers satisfactorily. The authors now include a description of the relative differences of solid angles between the two studies, which is good. However, the presentation is still somewhat confusing for readers, because the comparison of fractional apex angles and fractional solid angles introduces a huge difference simply due to the fact of switching from a one-dimensional to a two-dimensional/squared parameter (the solid angle is proportional to the surface area of the sphere). Thus, the effect of the optical correction is partially masked by the effect of this parameter conversion, and the current text fails to make this transparent and resolve it for the reader.

1) Without optical correction, the absolute difference of 70.3° and 21.7° corresponds to solid angles of 1.15 sr and 0.112 sr, which is a 10-fold relative difference [according to the equation Ω=2π1-cosθ; Ω corresponds to the solid angle, 2θ to the apex, see https://en.wikipedia.org/wiki/Solid_angle]. After optical correction, the relative difference is reduced to 3-fold (0.24 sr vs. 0.08 sr).

Our suggestion is to present results in the main text and in Figure 2C in relative steradian terms. The authors could state that the initial 10-fold difference in covered stimulus area (between the two studies) was surprising, but that after optical correction, this difference is reduced markedly (by a factor of 3).

The remaining 3-fold difference could potentially be explained by a dependence of behavior on the spatial stimulus location (as the authors already write). If the authors structure the paragraph like this, they could then also delete the following sentence, which they need in their current version to present results accurately, but which can be confusing and anticlimactic for readers and the story: "These corrections did not eliminate the discrepancy in relative terms, as Dunn et al. still found a critical size that was approximately three times as large as Temizer et al.".

2) We are not convinced that a 50.3° difference is "far more striking" than a similar relative difference at small stimulus sizes (as the authors state in their point-by-point response). Sensory systems typically encode stimulus magnitudes in logarithmic terms (Weber-Fechner law).

According to the logic used by the authors, a difference of 50.3° is striking irrespective of the base stimulus size. I think they would agree though, that two stimuli, 360° and 310.7° in size, are not very different. This is why the description of differences in relative terms is important, which the authors have now included in their revision.

---

## [Author Response]

Reviewer #1:[…] 1) They calculate, how stimuli have likely been distorted in two recent publications. This is a very helpful type of analysis that can potentially correct/explain observed differences in studies. The authors state that in the original publications the critical stimulus sizes were reported as 21.7° and 72.0° (difference: factor 3.3), and after their correction, the stimulus sizes were 0.6% and 1.9% (of 4π steradians, difference: factor 3.2). If I understand the authors correctly, the authors take the small absolute difference of 1.9% and 0.6% ( = 1.3%) as evidence for a better match after optical correction, because the absolute difference (14%) in the original studies before correction (72° – 21.7°=50.3° ~ = 14% of 360°) had been much bigger. However, I don't agree that this is the best way to compare the results. The relative difference is about a factor of 3 in both cases, so not much different after correction. That being said, it is of course nonetheless very useful to perform these distortion calculations as they can be used to report the stimulus characteristics from the animal's perspective, which is what matters.

We thank the reviewer for this comment. We have comprehensively revised the paragraph describing the looming results to hopefully make our findings and interpretations clearer. However, we continue to think that *absolute* differences are more relevant than *relative* differences in the current context. Our rationale here is two-fold. First, both prior zebrafish papers on this topic put forward the interpretation that zebrafish exhibit a behavioral escape response when the looming stimulus exceeds a critical size, in *absolute* terms. This notion of critical size is deeply engrained in the literature, following numerous studies on multiple animals, and we think that reporting the observed discrepancy in *absolute* terms is more concordant with the literature. Second, because zebrafish were hypothesized to escape when looming stimuli exceed a critical size, we assert that the *absolute* discrepancy of 50° is far more striking than if the *relative* discrepancy of 3 has resulted in a smaller *absolute* change. For example, if one group had measured a critical angle of 5° and the other had found a critical angle of 15°, then we would have been far less concerned about the discrepancy, and we may not have started to think about the effects of optical refraction on the experimental results. To make these rationales for using *absolute* discrepancies more explicit, we edited the manuscript to say:

“This large angular discrepancy of 50.3° initially raises doubt to the notion that the stimulus’s absolute size triggers the escape (Hatsopoulos et al., 1995; Gabbiani, Krapp and Laurent, 1999; Fotowat and Gabbiani, 2011. However, a major difference in experimental design is that Temizer et al. showed stimuli from the front through a curved air-water interface, and Dunn et al. showed stimuli from below through a flat air-water interface.”

We hope this will help future readers understand why we were surprised by, and interested in, the *absolute* discrepancy in the experimental results. Nevertheless, we agree with the reviewer that some readers may be interested in interpreting the discrepancy’s magnitude in *relative* terms, and we have edited the manuscript so that it now includes both methods of quantification. In particular, we added a sentence to the manuscript that reads:

“These corrections did not eliminate the discrepancy in relative terms, as Dunn et al. still found a critical size that was approximately three times as large as Temizer et al.”

This change to the manuscript will allow future readers to decide for themselves whether they want to think about the discrepancy in *absolute* or *relative* terms.

2) A programming tool is provided to help scientists calculate the distortions in their setups. It would be helpful if the functionality (inputs, outputs, available configurations) were described in more detail in the manuscript. The tool can be used for the experimental setup in Figure 1A, but for other configurations, users likely need to adapt it or use other software for ray tracing.

We thank the reviewer for correctly pointing out that we insufficiently described the tool in the manuscript. We have added a section titled “Computational tool for simulating optical distortions” to the “Materials and methods” of the paper. This new section reads:

“With this paper, we provide a computation tool for visualizing and correcting distortions (https://github.com/spoonsso/snell_tool/). […] Importantly, note that some stimuli will be physically impossible to correct (e.g. undistorted image content cannot be delivered outside the Snell window).”

Reviewer #2:[…] I have two main comments:1) The authors do not appear to have considered the angular intensity profiles of the screens used to present visual cues. However, this will influence the most eccentric location at which an image feature (pixel) is visible to the fish as well as the observed intensity distribution. Moreover, the types of screens used in different studies vary. For example, Dunn et al. used a diffusive screen (for back projection) which will have some scattering characteristic – how was this modeled? Temizer et al. used an OLED screen where individual pixels have a (likely more limited) angular emission profile. This will determine the maximal theta' from which a ray can be emitted that will reach the fish, causing clipping at eccentric locations an affecting the size and amplitude profile of receptive fields. Can the authors incorporate the angular intensity profile of the presentation screen into their modeling (tool)?

We thank the reviewer for this suggestion. The reviewer is correct that we did not explicitly consider the angular intensity profiles used to present visual cues, and we assumed that the apparatuses used by Dunn et al. and Temizer et al. uniformly emitted light over all angles. We agree that the OLED screen used by Temizer et al. likely had a “more limited” angular emission profile that could have contributed to differences between the two studies. Similar OLED screens will also be relevant to many users of the tool, and we agree that OLED monitors might have important consequences for receptive field mapping experiments. We therefore accepted the reviewer’s suggestion and changed our tool to allow the user to specify non-uniform angular emission profile. We clarified these points in our paper by adding a sentence to the Materials and methods that reads:

“Here we assumed that the virtual screen emits light uniformly at all angles, but this assumption is violated by certain displays, and our computational tool allows the user to specify alternate angular emission profiles.”

We also demonstrate this new functionality in a modified notebook that we provide alongside the computational tool. We think that these modifications will make the tool more useful for users.

However, two considerations lead us to conclude that detailed corrections for the angular emission profiles of Dunn et al. and Temizer et al. were not needed to quantitatively compare their results. First, Dunn et al. used a standard light diffuser that is near uniform over the modest range of angles needed to fully fill the Snell window and specify the size of the looming stimulus. Second, Temizer et al. only showed looming stimuli in front of the fish, so the OLEDs were only used in a narrow operating regime where their angular emission profile were also nearly uniform. The quantitative effects of non-uniform angular emission profiles are now illustrated in the tool’s modified notebook, including a demonstration that effects from OLED-like profiles are minor for small angles near the center of the Snell window.

2) It was not clear to me why the maximum image size is assumed to be 360° or 4π steradians. In the Dunn and Temizer papers, I assume angular size is computed according to a simple 2D model where one eye views a directly approaching orthogonal object.In this case, the largest angular size subtended at the eye is 180° (at impact). However, the largest image size cast on the retina of a real eye will be limited by the visual field of that eye (and thus occur prior to impact). Similarly, for the analysis of solid angle, the authors compute percentages assuming a maximal size of 4π, which is clearly not biologically plausible. I would find it more useful if image sizes were given as a proportion of the (monocular) visual field. What is this estimated to be?

We thank the reviewer for this comment. Some type of normalization is needed to compare critical size discrepancies expressed in different units (i.e. angle versus solid angle), but we agree that the normalization’s divisor is arbitrary. Given this arbitrariness, we decided that it was most straightforward to simply normalize each quantity by the maximal mathematical value that is possible for the quantity under consideration. This corresponds to 360° for angles and 4π steradians for solid angles. We agree that we could instead try to normalize by the estimated size of the fish’s visual field. However, we do not think that this normalization scheme would be any less arbitrary, and we think that the complication involved in computing it and justifying it would significantly decrease its utility. For example, what would the size of the visual field even mean when expressed in units of degrees?

On the other hand, we totally agree that the looming stimulus will never reach these maximal sizes, and we apologize that we were unclear in this regard. To make these normalization issues clearer for future readers, we have comprehensively revised the presentation of our looming results in light of this comment. We now initially present the discrepancies in the absolute units of degrees and steradians. We then introduce the normalization factor in an explicit way that makes its intended interpretation more explicit to the reader. In particular, we now say:

“This leaves a discrepancy of 0.16 steradians, which is a much smaller fraction of the full spherical solid angle (4π steradians) than 50.3° was of the full circular planar angle (360°). Correcting with Snell’s law thus markedly reduced this discrepancy in the literature, shrinking an original fractional discrepancy of 14.0% ± 1.6% to 1.3% [+0.6, -0.5]% (units normalized for comparison to 360° and 4π steradians, respectively) (Figure 2C).”

Reviewer #3:[…] Here are some comments I hope will make the paper even more enjoyable:1) Would it be possible for the authors to setup their repository to work with "My Binder" (https://mybinder.org/), so that readers who are not using/familiar with python could still use their notebook?

We have accepted the reviewer’s very good suggestion. In particular, the README for the tool now contains a My Binder link that we hope will make the tool useful to a wider audience. We have added a sentence to README file that says,

“Users can also run the tool without installing python by clicking on the `launch binder’ link above.”

2) In the second paragraph of the main text the authors state that in traditional experimental setups for zebrafish, due to Snell's law, stimuli that are distant appear to the fish at the asymptotic value of ~48.6°, and that this would lead to a "Snell window" of 97.2°. I think this leads to some interesting questions:2.1) In this setup configuration, the Snell window is covering what area of the bottom of the Petri dish? It would be nice to see visually represented what is the projection size of the window as size of the dish, since:2.1.1) If the window is only covering part of the dish, the animals are freely moving, and the correction algorithm is used, this would mean that as the animals move, they could go past the edge of the window and be in a region where the stimulus is not present?2.1.2) It would be nice to know how big the window projection is in relation to the animal's visual space, and how the distance between the petri dish and the projection window change this

The Snell window is relative to the fish, so it moves around as the animal explores its aquatic environment. As such, the fish cannot move beyond the edge of the Snell window. To make this point more clearly, we edited Figure 1C to add a small zebrafish at the center of the Snell window.

The reviewer is correct that the distance between the petri dish and the projection screen has a major impact on the visual distortions experienced by the fish. We have edited the legend to Figure 1C to state the size of the screen and to point out that different distances strongly affect the relative sizes of the screen and Snell window. In particular, the Figure 1C legend now says:

“The virtual screen is modeled as a 4 x 4 cm square with 250 pixels / cm resolution, and we fixed the total distance between the fish and the virtual screen, da+dw, to be 1 cm. Note that only a fraction of the screen is visible when da/dw is small (bottom left), but a distorted view of the full screen fits within the Snell window when da/dw becomes large (bottom right).”

3) The authors also mention that having water instead of air in the space between the petri-dish and the projection could be used as an alternative solution to the problem (main text, last paragraph), with the shortcoming of having a reduced transmission of stimuli in large angles. I wonder what these large angles would be? Would this be something easy to calculate? I wonder about this, since Franke et al. (https://elifesciences.org/articles/48779), show a "fish cinema" system where a lot of the fish's field of view is covered (while projecting directly at a screen which was the water container wall – so no air or water interface before the projected image)

We agree with the reviewer that one appealing way to alleviate optical distortions is to eliminate air-water interfaces altogether, either by using a diffuser that works well outside air, or by engineering the diffractive elements into the walls of the chamber itself. In principle, we think these approaches represent very good ideas. At the moment, we do not have a quantitative method to model how well a general light diffuser would work outside air, and we were thus unable to confidently gauge how well the setup in Franke et al. is working. We note that our own thinking on these issues has been informed by experimenting with wet diffusive screens (e.g. Rosco Cinegel) in a laboratory setting, and we encourage the reviewer to similarly experiment with their own setup to see how well light is spreading throughout the aquatic environment. We are sorry that we cannot say anything more definitive on this point, but we do not want to mislead the reviewer with speculative hypotheses. We have added a citation to Franke et al. as another example paper that engineered the diffuser into the fish tank.

4) One thing that I didn't see mentioned in the paper is chromatic aberration. Given the fact that stimuli are travelling through 3 different media at least, one could expect that different wavelengths will be projected slightly misaligned when compared to one another, especially considering the wide chromatic range present in Zebrafish vision? I wonder if it would be easy to incorporate these calculations in the tools they are describing, and most importantly, how could this affect the calculation of receptive fields, considering the misaligned patches of different wavelengths could directly influence some of these receptive fields?

The reviewer is correct that we did not consider chromatic effects in the initial submission, and we agree that chromatic aberration is quantitatively important and easy to treat with the tool. We have thus modified the tool’s notebook to point out how users can use the tool to consider chromatic effects. In particular, when defining the variable “n_w_,” we now say:

“This parameter varies with the wavelength of light and can be changed to simulate chromatic aberration.”

We thought that the reviewer might also be interested in the results of some numerical experiments that we performed to explore the effects of chromatic aberration. In particular, we simulated the effects of optical refraction on a distorted checkerboard pattern, which we designed to transform into an undistorted form through refraction (Author response image 1). When the projected pattern consists of equal parts red (n_w_ = 1.331) and blue (n_w_ = 1.341) light, the red and blue components of the transformed images combine to produce a dual-channel stimulus that is usually purple. However, chromatic aberrations did result in a few red or blue pixels at the edges of the checkerboard. Note that the reviewer might need to zoom in on Author response image 1 to see these relatively small effects.

**Author response image 1. respfig1:** Left: The image of a grayscale checkerboard that our tool suggests a user display in order to achieve an undistorted checkerboard from the point of view of the fish. Center: Simulation of chromatic aberrations for the resulting transformed checkerboard when it consists of equal parts red and blue light. Right: Zoom of image at Center.

Also note that the size of the Snell window depends slightly on the wavelength of light, which may lead to additional interesting aberration effects.

[Editors' note: further revisions were suggested prior to acceptance, as described below.]

The manuscript has been improved but there are some remaining issues that need to be addressed before acceptance, as outlined below:The reviewers have addressed most of the comments from the three reviewers satisfactorily. The authors now include a description of the relative differences of solid angles between the two studies, which is good. However, the presentation is still somewhat confusing for readers, because the comparison of fractional apex angles and fractional solid angles introduces a huge difference simply due to the fact of switching from a one-dimensional to a two-dimensional/squared parameter (the solid angle is proportional to the surface area of the sphere). Thus, the effect of the optical correction is partially masked by the effect of this parameter conversion, and the current text fails to make this transparent and resolve it for the reader.

We are glad that the reviewers were satisfied by the bulk of our first revision. As suggested, the current revision further improves the presentation of the looming results.

1) Without optical correction, the absolute difference of 70.3° and 21.7° corresponds to solid angles of 1.15 sr and 0.112 sr, which is a 10-fold relative difference [according to the equation Ω=2π1-cosθ; Ω corresponds to the solid angle, 2θ to the apex, see https://en.wikipedia.org/wiki/Solid_angle]. After optical correction, the relative difference is reduced to 3-fold (0.24 sr vs. 0.08 sr).

We agree that quantifying the size of the looming stimuli with their solid angle increases the relative difference between the two reported values of critical size. We also agree that “the effect of the optical correction is partially masked by the effect of this parameter conversion, and the current text fails to make this transparent and resolve it for the reader.” However, the formula provided by the reviewers does not apply the stimuli presented by Dunn et al. As such, the relative size difference is slightly smaller than suggested by the calculation above. We find 9.0-fold with our method, whereas 1.15/0.112 = 10.2. The reviewers’ basic point still stands.

Our suggestion is to present results in the main text and in Figure 2C in relative steradian terms. The authors could state that the initial 10-fold difference in covered stimulus area (between the two studies) was surprising, but that after optical correction, this difference is reduced markedly (by a factor of 3).

We thank the reviewers for this good suggestion. We have accepted it and added the relative size comparison in Figure 2C. In the text, we now set up the problem by saying:

“Although Temizer et al., 2015 and Dunn et al., 2016, both found that a critical size of looming stimuli triggered escape behavior, they reported surprisingly different values for the critical angular size (21.7° ± 4.9° and 72.0° ± 2.5°, respectively, mean ± 95% CI). […] This large size discrepancy initially raises doubt to the notion that a stimulus size threshold triggers the escape (Hatsopoulos et al., 1995; Gabbiani, Krapp and Laurent, 1999; Fotowat and Gabbiani, 2011.”

We then resolve the problem by performing the Snell’s law corrections and saying:

“This leaves a discrepancy of 0.16 steradians, which is much smaller than the original solid angle discrepancy of 0.91 steradians (Figure 2C, black). […] The small remaining differences could indicate an ethologically interesting dependence of behavior on the spatial location of the looming stimulus (Dunn et al., 2016; Temizer et al., 2015.”

The remaining 3-fold difference could potentially be explained by a dependence of behavior on the spatial stimulus location (as the authors already write). If the authors structure the paragraph like this, they could then also delete the following sentence, which they need in their current version to present results accurately, but which can be confusing and anticlimactic for readers and the story: "These corrections did not eliminate the discrepancy in relative terms, as Dunn et al. still found a critical size that was approximately three times as large as Temizer et al.".

We agree that the suggested changes improve the manuscript and alleviate the need for this sentence. We have eliminated it from the manuscript.

2) We are not convinced that a 50.3° difference is "far more striking" than a similar relative difference at small stimulus sizes (as the authors state in their point-by-point response). Sensory systems typically encode stimulus magnitudes in logarithmic terms (Weber-Fechner law).According to the logic used by the authors, a difference of 50.3° is striking irrespective of the base stimulus size. I think they would agree though, that two stimuli, 360° and 310.7° in size, are not very different. This is why the description of differences in relative terms is important, which the authors have now included in their revision.

We do not believe there are substantial disagreements here. We think that both absolute and relative differences contribute to whether or not a discrepancy seems “striking.” The reviewers’ appreciation for the merits of relative comparisons is clear from their above comment. Their appreciation that absolute comparisons remain valuable is implicit. We hope that we’ve now converged on phrasing that the reviewers will find satisfactory. We personally find the new phrasing much more elegant, and we thank the reviewers for pointing out various difficulties emerging from past wordings.